# A Systematic Review of Population-Based Studies Assessing Knowledge, Attitudes, Acceptance, and Hesitancy of Pregnant and Breastfeeding Women towards the COVID-19 Vaccine

**DOI:** 10.3390/vaccines11081289

**Published:** 2023-07-27

**Authors:** Vincenza Gianfredi, Pasquale Stefanizzi, Alessandro Berti, Marilena D’Amico, Viola De Lorenzo, Antonio Di Lorenzo, Lorenza Moscara, Silvana Castaldi

**Affiliations:** 1Department of Biomedical Sciences for Health, University of Milan, Via Pascal, 36, 20133 Milan, Italy; vincenza.gianfredi@unimi.it (V.G.); alessandro.berti@unimi.it (A.B.); marilena.damico@unimi.it (M.D.); viola.delorenzo@unimi.it (V.D.L.); silvana.castaldi@unimi.it (S.C.); 2Interdisciplinary Department of Medicine, University of Bari Aldo Moro, Piazza G. Cesare 11, 70121 Bari, Italy; antoniodilorenzo95@gmail.com (A.D.L.); lorenza.moscara@gmail.com (L.M.); 3Fondazione IRCCS Ca’ Granda Ospedale Maggiore Policlinico, Via Francesco Sforza, 35, 20122 Milan, Italy

**Keywords:** pregnant women, lactating, breastfeeding, COVID-19 vaccine, acceptance, knowledge, attitude, hesitancy

## Abstract

The anti-SARS-CoV-2 vaccination is recommended for pregnant women due to the high risk of complications. However, pregnancy has been associated with vaccine hesitancy. Our review aims at summarizing the existing literature about anti-SARS-CoV-2 vaccine hesitancy in pregnant and lactating women. The research was conducted on PubMed/MEDLINE, ExcerptaMedica Database (EMBASE), and Scopus, according to PRISMA guidelines. Articles regarding the COVID-19 vaccine’s acceptance and/or refusal by pregnant and lactating women were selected. Only observational, population-based studies were included. The Joanna Briggs Institute quality assessment tools were employed. A total of 496 articles were retrieved, and after the selection process, 21 papers were included in the current analysis. All the included studies were cross-sectional, mostly from Europe and North America. The sample sizes ranged between 72 and 25,111 subjects. All of them included pregnant subjects, except one that focused on breastfeeding women only. Vaccine hesitancy rates ranged from 26% to 57% among different studies. Fear of adverse events and lack of knowledge were shown to be the main drivers of hesitancy. Approximately half of the studies (11/21) were classified as low quality, the remaining (9/21) were classified as moderate, and only one study was classified as high quality. Primigravidae were also shown to be more likely to accept anti-SARS-CoV-2 vaccination. Our findings confirm significant anti-SARS-CoV-2 vaccine hesitancy among pregnant women. Information gaps should be addressed to contain concerns related to adverse events.

## 1. Introduction

Vaccine acceptance is influenced by multiple determinants dynamically interacting with each other, including the subject’s knowledge and attitudes regarding vaccination, societal norms, and perceived vaccination-related benefits and risks [1]. In case of insufficient drive towards vaccination, the subject might hesitate or even refuse vaccination. This wide spectrum of behaviors leading to delays in vaccination schedules’ completion is currently defined as vaccine hesitancy [2]. Hesitancy may arise from multiple factors, including concerns about the product’s safety and effectiveness, or mistrust in either vaccine development or regulatory processes. These attitudes and beliefs may stem from various causes, among which misinformation is one of the main determinants [3,4]. Vaccine hesitancy has been listed as one of the top ten threats to global health by the World Health Organization (WHO) due to its impact on vaccination coverage, leading to a decrease in immunization rates [5].

The complexity of these phenomena increases when dealing with pregnant/breastfeeding women [6]. In fact, during pregnancy and breastfeeding, women often seek information regarding their child’s health as well as their own, which is expected to significantly influence the subject’s behavior towards medical matters [7]. At the same time, these women are highly recommended to promptly undergo vaccination, as various vaccine-preventable diseases have a significantly higher chance of causing severe outcomes during pregnancy or might be spread to the child either before or after birth.

In this regard, the COVID-19 pandemic has strongly increased the sense of disorientation towards health choices and vaccination [8]. During the early months of the pandemic, neither an effective therapy nor a safe and effective vaccine were available, and significant expectations built up in the general population for a vaccine to “restore normalcy” [9]. At the same time, when anti-SARS-CoV-2 vaccines became available, the novelty of the type of vaccines used, as well as the initial paucity of data about long-term effects (safety and efficacy), engendered ambiguous feelings towards vaccination.

Pregnant and breastfeeding women, in particular, were concerned about possible side effects of the vaccine affecting either them or their child [10]. However, subsequent data from the Centers for Disease Control and Prevention suggest that there has been no significant increase in side-effects or complications among pregnant women vaccinated against COVID-19 compared to the general population [11]. Based on that, public health programs have prioritized pregnant women for vaccination, as they are a high-risk group for COVID-19 infection and its complications.

In this review, we aimed to assess and synthesize the existing literature on knowledge, beliefs, attitudes, barriers, and facilitators relating to acceptance/refusal of COVID-19 vaccination among pregnant and breastfeeding women. In particular, we aimed to answer the following research questions: (1) What is the level of knowledge regarding COVID-19 vaccination among pregnant/breastfeeding women? (2) What are the COVID-19 vaccine acceptancy facilitators and barriers associated with pregnancy and/or breastfeeding? We seek to provide a comprehensive understanding of the current landscape, identify research gaps, and highlight the implications for public health strategies aimed at promoting vaccination in this population. For homogeneity’s sake, we only addressed population-based studies in this review. Hospital based studies will be assessed in a companion paper, as women who actively look for medical support might differ significantly in terms of health literacy, attitude towards healthcare in general, and trust in medical professionals.

## 2. Materials and Methods

### 2.1. Search Strategy and Data Source

The current systematic review was conducted in accordance with the Cochrane Collaboration [12] and the results reported based on the Preferred Reporting Items for Systematic Reviews and Meta-Analyses (PRISMA) 2020 guidelines [13]. Details regarding search query are reported in Table 1.

Free text words and MeSH terms have been combined using Boolean operators and logically combined to build the appropriate search strategy. The full search strategy, adopted for each database, is available in Appendix A. Moreover, reference lists of included articles were also screened in order to detect any additional relevant articles. Finally, experts in the field were consulted in order to identify any supplemental articles not previously retrieved. The protocol of this review has been previously published [14].

### 2.2. Inclusion/Exclusion Criteria

To be included, the studies were required to meet the following inclusion criteria: (i) being an original, observational, population-based study (including cross-sectional, case-control, or cohort both prospective and retrospective studies); (ii) written in English; (iii) published in peer-reviewed international scientific journals; (iv) conducted after 2019 (because COVID-19 pandemic occurred at the end of 2019); and (v) focusing on knowledge, attitudes and practice of pregnant or breastfeeding women in taking/refusing COVID-19 vaccination. A detailed description of the inclusion/exclusion criteria is reported in Table 1.

### 2.3. Selection Process

All the retrieved studies were subsequently downloaded to the EndNote software (EndNote^®^ for Microsoft, Redmond, WA, USA, 2020). Duplicates were removed using an automatic function in the EndNote software, followed by a manual check by one of the authors. The remaining articles were then assessed for eligibility, firstly based on the title and abstract, followed by their full text.

### 2.4. Data Extraction

The data extraction process was performed in duplicate by two reviewer authors. A standardized and pre-defined Excel (Microsoft Excel^®^ for Microsoft 365 MSO, Redmond, WA, USA, 2019) spreadsheet was used to extract data from the included studies. The following information was extracted from each article: author name, study period, country where the study was conducted, study settings, main characteristics, and the study population’s number, study completion rates (attrition), tool(s) used to assess the outcomes, number of items, whether the tool(s) was/were validated or not, manner in which the questionnaire was administered, recruitment methods, outcomes of interest, outcomes definition, main results, funds and conflicts of interests, if any. Vaccine coverage was also recorded, if available. We also extracted methodological information, such as whether the tool was validated or not and the statistical analysis undertaken. Lastly, if studies reported data using risk estimates—for instance, odds ratio (OR), risk ratio (RR), or hazard ratio (HR)—the maximally adjusted data along with the list of variables used for the adjustment were recorded.

### 2.5. Quality Assessment

The risk of bias was independently assessed by two Authors, using the Joanna Briggs Institute (JBI) quality assessment tools [15]. For each of the 8 items, the score range between −2 and 2 points. Therefore, the total score could range between −16 and 16. Articles scoring from −16 to 4 will be classified as low quality, articles scoring from 5 to 9 as moderate, and articles scoring equal to or more than 10 (and up to 16) as high quality.

## 3. Results

### 3.1. Literature Search

A total of 496 articles were retrieved, of which 189 were from PubMed/Medline, 180 from Scopus, and 127 from EMBASE. After a preliminary screening, 94 articles were excluded because they were duplicates, 5 articles were removed for being written in a language other than English, and 330 articles were excluded based on title and abstract, remaining a total of 67 eligible articles. However, three papers were removed after full text assessment [16,17,18]. A detailed description of the excluded reasons is reported in Appendix A. At the end of the screening process, a total of 64 articles were considered included. However, due to the high heterogeneity in terms of outcomes assessed and its definition as well as study setting, we decided to present in the current manuscript only population-based studies (a total of 21 included studies) [19,20,21,22,23,24,25,26,27,28,29,30,31,32,33,34,35,36,37,38,39]. Consultation with experts did not add any further eligible studies. The full selection process is depicted in detail in Figure 1.

### 3.2. Main Characteristics of Included Studies and Quality Assessment

Main characteristics of included studies are shown in Table 2, listed in alphabetical order. In particular, the first two studies were conducted in April–June 2020, of which one was a European multicenter study [21], whereas the other was conducted in the United States of America [28]. Generally speaking, studies were conducted in almost all the continents, except Oceania; however, America and Europe were the geographical areas most frequently explored, with 7 and 6 studies, respectively. All the included studies were cross-sectional, even the study conducted by Mhereeg et al. [26] that performed a cross-sectional analysis starting from a cohort study. All the studies involved the administration of a questionnaire at least to explore the exposure(s) or the outcome(s) (for further details about the outcome, please see Appendix A). The majority of the studies were nation-wide, with the exception of two European-level studies [21,37] and four sub-national studies [20,26,34,39]. Participants were mainly (17/21) recruited via social media or posting the online survey link on specific websites. In some cases, posters redirecting to the survey link [26] or healthcare professionals (as for instance gynecologists or midwives) [19,26] were also involved in recruiting participants, which self-administered the questionnaire. Only three studies recruited and administered the survey via telephone [25,31,34]. While one study recruitment method was not explicitly reported, survey administration was performed face-to-face [20]. Most of the included studies (11/21) stated that they received funding to conduct their research activities; some of them (7/11) did not receive any funding, whereas three studies did not report this information [20,22,29]. Lastly, all the studies stated that they did not have conflicts of interests, except one that did not report the information [29]. Details regarding the outcome of interest, including its definition, tool(s) used with number of items, and validation status are reported in Appendix A.

### 3.3. Quality Assessment

Regarding the quality assessment, approximately half of the studies (11/21) scored equal or below 4 and therefore were classified as low quality, the remaining (9/21) scored between 5 and 9 and therefore were classified as moderate, and only one study scored 12 and therefore was classified as high quality [25]. The overall score and the overall appraisal of quality is reported in Figure 2. Appendix A reports the item-by-item quality assessment for each included study. Inter-rater reliability was assessed, and discrepancy among the two reviewers was 20%. Disagreements were solved through discussion and a final agreement was reached for all the included studies.

### 3.4. Characteristics of the Studied Population

Main characteristics of studied population are reported in Table 3. In brief, approximately half of the included studies (11/21) included both pregnant and breastfeeding women, the remaining half of the studies (9/21) only included pregnant women, while only one study included breastfeeding women [39]. The age of included women ranged between 18 and 55 years. The smallest sample size reported was 72 subjects [25], whereas the largest was 25,111 subjects [26]. Finally, approximately half of the studies (10/21) did not report a completion rate, and all the remaining competition rates ranged between 63% [29] and 100% [23].

### 3.5. Attitude toward COVID-19 Vaccine

Nine studies explored attitude toward COVID-19 vaccination and its determinants (Table 4). Two studies assessed attitude toward COVID-19 booster doses [38,39] and its determinant. The positive attitude toward COVID-19 vaccine ranged between 29.7% [37] and 80% [28] among pregnant women, and between 38.6% [37] and 83.4% among breastfeeding women [28]. However, one study did not used a composite measure but assessed attitude toward specific aspect of the COVID-19 vaccine [31]. Finally, one study expressed attitude using a continuous scale [19].

Regarding predictors, many factors were considered, including the following: (i) sociodemographic factors such as age, ethnicity, educational level, marital status, living with 65-year-old family member, income, and work inactivity; (ii) gestational characteristics such as gestational week, gestational diabetes, being primigravida, and breastfeeding; (iii) vaccination behavior, including having received vaccine recommendation during pregnancy, having received influenza or tetanus vaccine, and fear about COVID-19 vaccine; (iv) COVID-19 related aspects, including testing positive, sources consulted regarding COVID-19, anxiety and fear of COVID-19, and the impact of COVID-19 pandemic on life (Appendix A).

#### 3.5.1. Sociodemographic Factors and Attitude toward COVID-19 Vaccine

Regarding age, only one study [37] out of two studies [36,37] found a statistically significant association between women older than 40 years old and a higher attitude toward vaccination [37]. Ethnicity was variably assessed among three studies; nevertheless, regardless of the categories used to define ethnicity, minorities were significantly associated with a higher attitude [28,36,37]. Educational level was assessed in three studies [21,22,37], which showed that a higher educational level was associated with a higher attitude [22,37]. However, medium educational level was associated with a lower attitude [21]. Having a partner [19] and living with a 65-year-old family member [37] were only explored in one study each, and only having a partner was associated with higher attitude toward COVID-19 vaccine. Income was assessed in three different studies, but the results were discordant. Actually, lower income was associated both with lower [19] and higher attitude [22], whereas higher income was associated with higher attitude [22]. Lastly, only one study assessed the association between professional inactivity and lower attitude [21] (Appendix A).

#### 3.5.2. Gestational Characteristics and Attitude toward COVID-19 Vaccine

Three studies out of eleven assessed the association between selected gestational characteristics and attitude toward COVID-19 vaccine. In particular, being primigravida was associated with a higher attitude in two studies [21,28], as well as having gestational diabetes [22]. On the contrary, gestational week was associated with a lower attitude in one study. Moreover, among women who gave birth within one year, breastfeeding was associated with higher attitude [21] (Appendix A).

#### 3.5.3. Previous Vaccination Behavior and Attitude toward COVID-19 Vaccine

Having received influenza vaccination in the past [28], as well as receiving recommendations during pregnancy to get influenza vaccination [22,37], were both associated with higher attitude toward COVID-19 vaccine. Similarly, being vaccinated against tetanus during the current pregnancy was also associated with a higher attitude [22]. On the contrary, being afraid of potential side effects to the baby from the COVID-19 vaccine was associated with a lower attitude toward vaccination [22] (Appendix A).

#### 3.5.4. COVID-19 Related Aspects and Attitude toward COVID-19 Vaccine

Having tested positive to COVID-19 was explored only in one study, but no statistically significant association was found with attitude [37]. Moreover, the sources used to retrieve information on COVID-19 were also explored only in one study, and data showed a positive association between using official data sources and a positive attitude [19]. Furthermore, the negative impact of COVID-19 on life was also assessed in one study, but no statistically significant association was found (Appendix A).

#### 3.5.5. Attitude toward COVID-19 Booster Doses

Only two studies assessed the attitude toward COVID-19 booster doses, and they found that approximately 57% of pregnant women had a positive attitude toward booster doses. One study assessed the negative attitude, while the second study assessed the positive attitude. In the two studies, several determinants were associated with negative/positive attitude. Specifically, higher levels of education, and higher anxiety and depression scores were associated with a lower negative attitude. On the other hand, mixed feeding, longer breastfeeding duration, a history of high-risk travel, and better physical health were all associated with higher negative attitude [39]. Regarding the positive attitude, income and residency were associated with a lower positive attitude, while level of education, previous infection, having received influenza vaccine, worries about infection, and commitment to immunize children were all associated with a higher attitude toward the COVID-19 vaccine [38] (Appendix A).

### 3.6. COVID-19 Vaccine Acceptance

COVID-19 vaccine acceptance was investigated in nine studies [23,24,25,26,29,30,31,32,39], with acceptance rates ranging from approximately 20% [39] to 68% [26] (Table 4). The most frequently reported reasons for accepting the COVID-19 vaccine were to protect mother and baby, the perceived beneficial effect, and satisfactory research on safety [26].

Moreover, several factors were found to be associated with COVID-19 vaccine acceptance. In particular, being older than 30 years [24,26], ethnicity (Asian [26], black/Africans ethnicity [29], and in general minorities [24]), higher income [26,29], higher education [24], having received the influenza vaccine [32], or all recommended vaccines [24], having received healthcare provider recommendation [30,32], were all statistically significant associated with higher acceptance. Moreover, higher knowledge on COVID-19 vaccine was associated with higher acceptance rate [30]; on the contrary, fear of side effects both for women and newborns and perceived barriers for vaccination [24] were associated with lower acceptance rate [30]. Lastly, the perceived threat of COVID-19 [29], perceived benefits of the COVID-19 vaccine [24], behavioral attitudes toward infection mitigation, and living with a higher-risk subjects [32] were also statistically significant associated with higher acceptance. A significant correlation was found between negative attitude and anxiety of COVID-19 (r = 0.02 [23]) and between negative attitude and fear of COVID-19 [23] (Appendix A).

### 3.7. COVID-19 Vaccine Hesitancy

Vaccine hesitancy was assessed in five studies [27,32,33,34,35] and ranged between 26% [27] and 57% [35] of the sample (Table 4). In one study [34], hesitancy was assessed using a continuous scale. The most frequently reported reasons for hesitancy were concerns about safety of the vaccine for pregnant/breastfeeding women or their baby [32,33], the perception of a rapid vaccine development and approval process [32], and a lack of knowledge/information about the vaccine [33,35]. Looking at predictors of hesitancy, concerns over vaccine safety [27], the belief in the superiority of natural immunity (lasting longer/better/safer) [27], no previous tetanus vaccination [34], and lower income [34] were all significantly associated with higher level of hesitancy. Conversely, lower educational level [34], younger age [35], primiparity [35], living in a less urban area [35], and higher knowledge on vaccine [35] were all associated with lower level of hesitancy. Having a chronic disease [34], receiving vaccine advice [34], and ethnicity [35] were not found to be associated [34] with hesitancy (Appendix A).

### 3.8. Fear Related to COVID-19 Vaccination

The reasons for fear of the COVID-19 vaccine were evaluated in seven studies, out of which only two also investigated potential associate factors (Table 4). Specifically, seven studies assessed concerns regarding safety and fear of side effects. In particular, five of them assessed concerns regarding safety/fear of side effects in general, of which three studies [19,34,36] reported the frequency of this perceived fear, ranging between 3.9.7% [19] and 62% [34], whereas two studies expressed it on a continuous scale [23,25]. Safety concerns of the COVID-19 vaccine during pregnancy were assessed in two studies [19,20], with approximately 30% of the sample reporting this fear. The perception of potential harm to babies from the COVID-19 vaccine was investigated in one study [39], which reported that 74.4% of the participants interviewed expressed this concern [39]. Another aspect of fear explored was receiving the COVID-19 vaccine in relation to trust in companies and data. In particular, Sezerol et al. [34] found that 18% of the sample considered the COVID-19 vaccine unnecessary, while approximately 13% did not trust the manufacturing companies. Skirrow et al. [36], on the other hand, discovered that 46% of the sample experienced fear of the COVID-19 vaccine due to a lack of safety data, and 17% expressed concerns about the speed of vaccine development. Lastly, two studies assess fear of the COVID-19 vaccine. The two studies that also explored predictors of fear found significant associations between fear and several factors. Negative attitudes [20], unplanned pregnancy [20], greater distance form healthcare facilities [20], lack of antenatal care utilization [20], lower educational level [20], and higher score on the vaccine conspiracy beliefs scale [25] were all significantly associated with higher levels of fear. However, fear was not associated with COVID-19 knowledge and parity (Appendix A).

### 3.9. Level of Knowledge on COVID-19 Vaccine

A total of three studies assessed the level of knowledge related to the COVID-19 vaccine (Table 4). Two of these studies used continuous scales to assess level of knowledge among both vaccine acceptant and refusal [25,39], while one study reported the frequency of participants with inadequate level of knowledge [20]. Due to the different scales used to assess level of knowledge across the studies, a direct comparison of results cannot be performed. However, in general, the studies indicated a significantly higher level of knowledge among individuals who accepted the COVID-19 vaccine compared to those who refused it. Moreover, over 50% of the participants had an inadequate level of knowledge [20]. Lastly, Mattocks et al. [25] also found that a higher level of knowledge significantly predicted the receipt of the COVID-19 vaccine (Appendix A).

## 4. Discussion

All available vaccines, apart from those based on live attenuate microorganisms for which pregnancy represents a temporary contraindication, showed acceptable safety and effectiveness during pregnancy. However, despite consistent evidence regarding the safety and effectiveness of vaccination during pregnancy and breastfeeding, vaccine hesitancy in pregnant and breastfeeding subjects is still a significant issue [6,40]. According to our review, COVID-19 vaccine hesitancy ranged between 26% and 57%. The most frequently reported reasons for hesitancy among pregnant and breast-feeding women were concerns about safety of the vaccine for pregnant/breastfeeding women or their baby [32,33], the perception of a rapid vaccine development and approval process [32], and a lack of knowledge/information about the vaccine [33,35]. As a matter of fact, healthcare professionals’ trust in vaccines and their recommendations to people is of paramount importance in order to increase vaccine acceptance. In the current review, we found that having received healthcare professional’s recommendations increased positive attitude toward COVID-19 vaccination, as well as on COVID-19 acceptance and in reducing COVID-19 vaccine hesitancy. In this respect, this source of data is another pivotal predictor of attitude toward vaccination. According to our review, official sources used to retrieve information on COVID-19 vaccine were associated with a positive attitude [19]. On the contrary, women who rely on non-scientific social media for vaccine-related information are often characterized by higher levels of anxiety regarding both infection and vaccine risks [41,42,43,44].

Anti-SARS-CoV-2 vaccination’s acceptance appears to be a very volatile variable [28,37], changing within a large range when different populations are taken into consideration. A higher level of education correlated with greater vaccination acceptance [22,37]. This could be related to an inability of women with lower formal education to fully understand the risks and benefits coming from immunization practices. Moreover, our review identified an inadequate level of knowledge regarding anti-SARS-CoV-2 vaccination for pregnant women [20], confirming that the ability to select and comprehend valid information sources is fundamental in determining the subject’s attitude towards vaccination. It is however interesting how various studies described higher acceptance for the anti-SARS-CoV-2 vaccine in minorities [28,36,37]. In fact, other papers which did not focus on pregnant subjects identified ethnic minorities and rural population as groups at a higher risk of vaccine hesitancy and refusal [45,46,47]. This difference might be related to pregnancy itself, which may represent a first contact occasion for immigrants and other groups who are socioeconomically disadvantaged, who are more prone to entrust their own health to healthcare professionals. The increased vaccine acceptance in primigravida women observed by some of the included studies [21,28] would suggest that this assumption has at least some degree of truth to it [28].

### 4.1. Implications for Policies and Practice

When considering public health policies and practice, this study suggests that even if vaccine hesitancy is mainly related to vaccine safety, healthcare providers’ uncertainty regarding the opportunity of offering vaccination to pregnant and breastfeeding women might largely influence vaccination attitude and acceptance. Consequently, taking every possible opportunity to offer vaccination to pregnant women is of utmost importance [48], considering that they are at an increased risk of severe outcomes for several vaccine-preventable diseases, including fetal anomalies, abortion, and maternal complications [49]. Nevertheless, vaccines are biological products, and as such must be treated with special attention [50]. In fact, pregnant and breastfeeding women are usually not included in trials, especially at the begging of the experimentation phase. Because of that and considering potential long term or rare adverse events following vaccination, trials assessing vaccine safety are not enough, but active surveillance programs and causality assessment approach should be further implemented [51,52]. Therefore, numerous studies have been carried out over time to verify the safety profile of commonly employed vaccines in pregnant women, which were shown to be safe for both the woman herself and the baby [53,54,55]. Moreover, pregnancy is a unique period in an individual’s life. Apart from the significant changes in the shape and physiology of her body, the expectant individual is subjected to emotional stress stemming from both changes in hormones’ incretion and psychological, social, and cultural factors [56]. Proper information and risk communication abilities should be acquired by healthcare professionals [57]. Indeed, transparent, up-to-date, trustful, and timely communication is necessary in order to address population’s fears and doubts and to spread correct information [58,59]. Pregnant women are actively seeking for health information both for themselves and their newborn [60]. Therefore, they can be more prone to consult several data sources, increasing the risk to be exposed to untrustworthy sources. Exposure to spectacularized information and data, often provided by sources whose reliability is hard to determine, may aggravate these subjects’ emotional status, leading to a misjudgment of the vaccination benefits and risks and, finally, to vaccination hesitancy or straight-up refusal [60,61]. Such issue constitutes a significant liability for healthcare systems worldwide: apart from increasing vaccination hesitancy, it is a form of self-exacerbating anxiety which might determine pregnancy complications especially in psychologically or physically vulnerable individuals [62]. Our study confirms that fear of adverse events is both a common occurrence [19,20,34,36,39] and one of the main drivers of anti-SARS-CoV-2 vaccine hesitancy in pregnant women [32,33].

In light of the above, communication techniques to healthcare professionals, of the current and future generation, should be a top priority for the entire healthcare systems [63,64,65]. Special attention should be dedicated to describing both risks and benefits of vaccination, as well as to reassuring the patient that they will be assisted for the whole duration of the pregnancy and promptly treated in case of any ill-effects [66,67]. In order to increase its efficacy, information should also be personalized, keeping each subject’s social and cultural background in mind [58,68]. Finally, healthcare professionals themselves should be both informed regarding current guidelines for vaccination of pregnant women and encouraged to recommend necessary vaccines to their patients [69,70,71].

### 4.2. Limitations and Strengths

Before we generalize our results, some limitations should be considered. Firstly, this is a secondary study and therefore it brings the intrinsic limitation of each primary study included. More specifically, several different outcomes were considered with different definition used in each study. Additionally, many different types of tools were used to assess outcome and therefore heterogeneity might be around it. Moreover, many different exposures were considered, limiting the possibility to statistically pooling the results. In fact, even if the effect sizes from each original study were extracted, they could not be pooled because less than three studies assessed the association between same exposure and same outcome. Moreover, in most of the cases studies did not adjust for potential confounders, or they were not specifically stated. Among those studies that adjusted for confounders, many different variables were computed. All the above might potentially impacting the final result. Furthermore, differences among studies’ results might be due to the differences among the study population and the sampling methods; in most cases, the questionnaire link was shared on social networks, selecting participants who voluntarily would take part of the study and probably biasing the results. Lastly, all data were self-reporting, with a certain risk of recall or social desirability bias.

Another limitation is related to the type of studies retrieved. Indeed, all included studies were cross-sectional.

Nonetheless, the current systematic review has certain strengths. First, we followed the PRISMA guidelines which allow us to use a comprehensive approach. Moreover, three different databases were consulted in order to retrieve all eligible studies (more than the minimum required by guidelines).

## 5. Conclusions

To conclude, this review summarized data from 21 cross-sectional studies focusing on pregnant and breastfeeding women aged 15–55 years, with a good geographical representativeness. The positive attitude toward COVID-19 vaccine ranged between 29.7% and 80% and several factors were considered as predictors. Among them, higher educational level, being primigravida, and having received previous vaccinations or vaccines recommendations from health care professionals were associated with a higher attitude, while results on socioeconomic status are discordant. COVID-19 vaccine acceptance ranged between 20% and 68%. In this case, older age, ethnic minorities, higher income and education, and having received previous vaccines or vaccines’ recommendations were all statistically significant associated with higher acceptance. Despite available evidence of safety and effectiveness of the anti-SARS-CoV-2 vaccination in pregnant women, hesitancy remains, thus threatening vaccination campaigns’ efficacy all over the world. According to our review, COVID-19 vaccine hesitancy ranged between 26% and 57%. Vaccine hesitancy’s main drivers appear to be fear and lack of information, proving that proper communication between healthcare professionals and their patients is the best way to counter uncertainties and doubts related to vaccination. Improving knowledge and awareness is the first point to increase vaccine confidence, and the failure to adequately inform pregnant women exasperates the skepticism about vaccination. An effort should be made to invest in communication-related soft skills of healthcare professionals, while also reinforcing mass communication via both new and traditional media.

## Figures and Tables

**Figure 1 vaccines-11-01289-f001:**
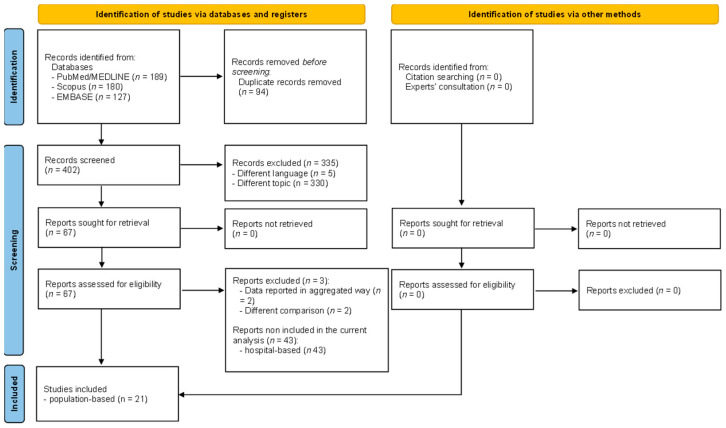
PRISMA flow diagram reporting the selection process.

**Figure 2 vaccines-11-01289-f002:**
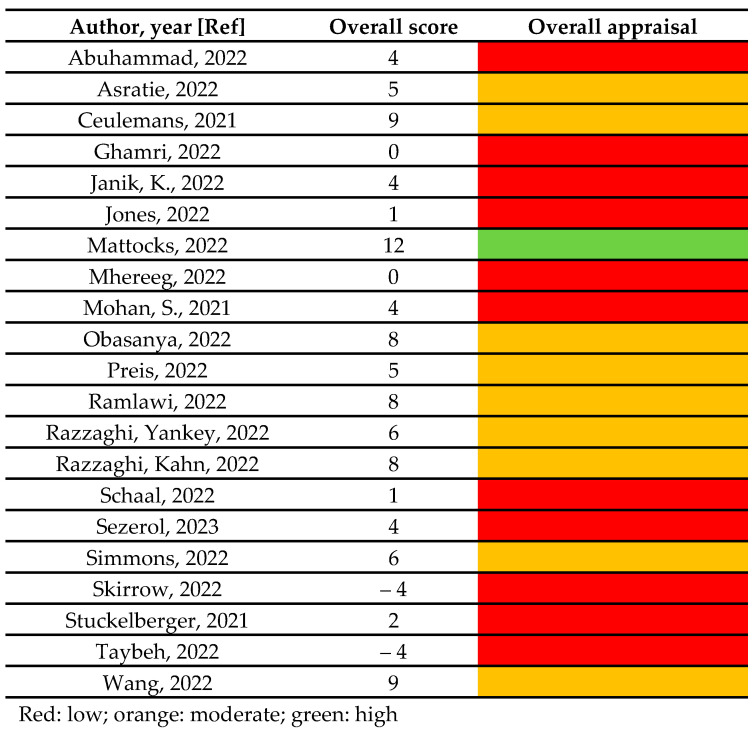
The overall score and the overall appraisal of quality [21,22,23,24,25,26,27,28,29,30,31,32,33,34,35,36,37,38,39,40,41].

**Table 1 vaccines-11-01289-t001:** Inclusion/exclusion criteria based on Population, Intervention/Exposure, Comparators/Controls, Outcome, Study design (PI/ECOS) strategy.

Search Strategy	Details
Search query	P: pregnant or breast-feeding women (and synonyms); E: COVID-19 vaccination (andsynonyms); O: knowledge, attitude, and practice (including factors associated with acceptance/hesitancy) regarding the COVID-19 vaccination (and synonyms).
Inclusion criteria	P: pregnant or breastfeeding women E: COVID-19 vaccination O: attitudes, acceptance, hesitancy, fear, knowledge S: original, observational study (including cross-sectional, case-control, or cohort both prospective and retrospective studies), published as peer-reviewed articles in international scientific journals
Exclusion criteria	P: studies not performed among humans or that were conducted on a different population as women in general, parents or only mothers of children older than one year, and children’s caregivers in general E: other than COVID-19 vaccination O: combining data with different and multiple outcomes, or assessing different outcomes not listed in our inclusion criteria (for instance, vaccine efficacy/safety/development or collecting serological/immunological data), and articles assessing acceptance/hesitancy/refusal against vaccines other than COVID-19 S: not original (reviews with or without meta-analysis), not performed among humans, not observational (as for instance trials), not published as peer-reviewed articles in international scientific journals (book, book chapter, thesis), no full-text papers (abstract, conference paper, letter, commentary, note)
Language	English
Time filter	After 2019
Databases searched	PubMed/MEDLINE, ExcerptaMedica Database (EMBASE) and Scopus
Search date	January 2023

**Table 2 vaccines-11-01289-t002:** Main characteristics of included studies, reported in alphabetical order.

Author, Year [Ref.]	Study Period	Study Design	Country	Study Setting	Recruitment Methods	Administration Method	Funds	CoI	QS
Abuhammad, 2022 [19]	September–October 2021	Cross-sectional	Jordan	nation-wide	social media, midwives, and gynecologists	self-administered	no	no	low
Asratie, 2022 [20]	December–February 2021	Cross-sectional	Ethiopia	community-based (Motta town and Hulet Eji Enese district)	n.a.	face-to-face	n.a.	no	moderate
Ceulemans, 2021 [21]	April–July 2020	Cross-sectional	BE, CH, IE, NL, NO, UK	multinational (6 European countries)	social media and websites dedicated to pregnant women	self-administered	yes	no	moderate
Ghamri, 2022 [22]	July–September 2021	Cross-sectional	Saudi-Arabia	nation-wide	social media	self-administered	n.a.	no	low
Janik, 2022 [23]	February–April 2022	Cross-sectional	Poland	nation-wide	social media dedicated to pregnant women	self-administered	yes	no	low
Jones, 2022 [24]	May–June 2021	Cross-sectional	USA	nation-wide	social media dedicated to pregnant women	self-administered	no	no	low
Mattocks, 2022 [25]	January–May 2021	Cross-sectional	USA	pregnant and postpartum veterans enrolled in VA care	mailed invitation followed by research telephone calls	telephone surveys (~45 min in length)	yes	no	high
Mhereeg, 2022 [26]	November 2021–March 2022	Mixed-method: cohort study (Databank) and cross-sectional	UK (Wales)	Born-In-Wales Birth Cohort	social media, and through midwives, and posters in hospitals	self-administered	yes	no	low
Mohan, 2021 [27]	October–November 2020	Cross-sectional	Qatar	nation-wide	HMC social media platforms	self-administered	no	no	low
Obasanya, 2022 [28]	April–June 2020	Cross-sectional	USA	nation-wide	Prolifc Academic	self-administered	yes	no	moderate
Preis, 2022 [29]	December 2020	Cross-sectional	USA	nation-wide	social media	self-administered	n.a	n.a.	moderate
Ramlawi, 2022 [30]	March–August 2021	Cross-sectional	Canada	nation-wide	Canadian social media accounts	self-administered	yes	no	moderate
Razzaghi, Yankey, 2022 [31]	April–November 2021	Cross-sectional	USA	nation-wide	NIS random-digit-dialing sample of cellular telephone numbers	household telephone survey	yes	no	moderate
Razzaghi, Kahn, 2022 [32]	March–April 2021	Cross-sectional	USA	nation-wide	internet panel operated by Dynata	self-administered	no	no	moderate
Schaal, 2022 [33]	March–April 2021	Cross-sectional	Germany	nation-wide	online platform	self-administered	yes	no	low
Sezerol, 2023 [34]	March–April 2022	Cross-sectional	Turkey	District Health Directorate in Sultanbeyli, district of Istanbul	telephone	via telephone	no	no	low
Simmons, 2022 [35]	December 2020–January 2021	Cross-sectional	USA (California)	nation-wide	StudyPages	self-administered	yes	no	moderate
Skirrow, 2022 [36]	August–October 2020	Cross-sectional	UK	nation-wide	social media, and telephone or Microsoft Teams interview	self-administered	yes	no	low
Stuckelberger, 2021 [37]	June–July 2020	Cross-sectional	Switzerland	European multi-centers	websites, forums, and social media	self-administered	no	no	low
Taybeh, 2022 [38]	November 2021–January 2022	Cross-sectional	Jordan	nation-wide	social media not otherwise specified	self-administered	no	no	low
Wang, 2022 [39]	September–December 2021	Cross-sectional	China	sub-national (southern China)	WeChat groups	self-administered	yes	no	moderate

BE: Belgium; CH: Switzerland; CoI: conflicts of interests; HMC: Hamad Medical Corporation; IE: Ireland; Ref: reference; n.a.: not available; NIS: National Immunization Survey; NISACM: National Immunization Survey Adult COVID Module; NL: Netherlands; NO: Norway; QS: quality score; UK: United Kingdom; USA: United States of America; VA: Veteran Affairs.

**Table 3 vaccines-11-01289-t003:** Main characteristics of studied population. Studies reported in alphabetical order.

Author, Year [Ref.]	Type of Population	Age in Years	Sample Size	Study Completion Rate
Abuhammad, 2022 [19]	pregnant and lactating	18–55	414 (pregnant 195, lactating 218)	82%
Asratie, 2022 [20]	pregnant	31.2 ± 5.76	851	n.a.
Ceulemans, 2021 [21]	pregnant and breastfeeding women up to three months postpartum	n.a.	16,063 (6661 pregnant, 9402 breastfeeding)	n.a.
Ghamri, 2022 [22]	pregnant	31.57 ± 7.79	5307	n.a.
Janik, K., 2022 [23]	pregnant	19–42	288	100%
Jones, 2022 [24]	pregnant and within six-months postpartum	29.61 (±3.89)	227	n.a.
Mattocks, 2022 [25]	pregnant	35.9 ± 2.7	72	71.3%
Mhereeg, 2022 [26]	pregnant	18–50	25,111	88.6%
Mohan, 2021 [27]	pregnant and lactating	18–46	341	n.a.
Obasanya, 2022 [28]	pregnant and post-partum	18–49	489	n.a.
Preis, 2022 [29]	pregnant	31.24 ± 4.24	1899	63%
Ramlawi, 2022 [30]	pregnant and lactating	30–39	3446	93.4%
Razzaghi, Yankey, 2022 [31]	pregnant and breastfeeding	18–49	7173 (3433 pregnant, 3740 breastfeeding)	n.a.
Razzaghi, Kahn, 2022 [32]	pregnant	18–49	1516	91.2%
Schaal, 2022 [33]	pregnant and breastfeeding	pregnant 31.8 ± 4.3 breastfeeding 32.4 ± 4.4	2339 women (1043 pregnant, 1296 breastfeeding)	n.a.
Sezerol, 2023 [34]	pregnant	28.07 ± 5.03	561	n.a.
Simmons, 2022 [35]	pregnant	18–45	387	86.2%
Skirrow, 2022 [36]	pregnant, breastfeeding	30–34	1181	77.4%
Stuckelberger, 2021 [37]	pregnant and breastfeeding	33	1551 (515 pregnant, 1036 breastfeeding)	75.1%
Taybeh, 2022 [38]	pregnant and lactating	29.7	584	n.a.
Wang, 2022 [39]	lactating	30.9 ± 4.8	432	85.4%

**Table 4 vaccines-11-01289-t004:** Overview of number and references assessing each specific topic. References for each topic are reported in alphabetical order.

Topic	N of Ref	Author, Year [Ref]
Attitude	8	Abuhammad, 2022 [21]; Asratie, 2022 [22]; Ceulemans, 2021 [23]; Ghamri, 2022 [24]; Obasanya, 2022 [30]; Razzaghi, Yankey, 2022 [33]; Skirrow, 2022 [38]; Stuckelberger, 2021 [39]
Attitude toward COVID-19 booster doses	2	Taybeh, 2022 [40]; Wang, 2022 [41]
Acceptance	9	Janik, K, 2022 [25]; Jones, 2022 [26] Mattocks, 2022 [27]; Mhereeg, 2022 [28]; Preis, 2022 [31]; Ramlawi, 2022 [32]; Razzaghi, Yankey, 2022 [33]; Razzaghi, Kahn, 2022 [34]; Wang, 2022 [41]
Hesitancy	5	Mohan, 2021 [29]; Razzaghi, Kahn, 2022 [34]; Schaal, 2022 [35]; Sezerol, 2023 [36]; Simmons, 2022 [37]
Fear	7	Abuhammad, 2022 [21]; Asratie, 2022 [22]; Janik, K., 2022 [25]; Mattocks, 2022 [27]; Sezerol, 2023 [36]; Skirrow, 2022 [38]; Wang, 2022 [41]
Knowledge	3	Asratie, 2022 [22]; Mattocks, 2022 [27]; Wang, 2022 [41]

N: number; Ref: reference.

## Data Availability

Upon request.

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
