# Peer review of "A Systematic Review of Population-Based Studies Assessing Knowledge, Attitudes, Acceptance, and Hesitancy of Pregnant and Breastfeeding Women towards the COVID-19 Vaccine"

_vaccines, 2023, doi:10.3390/vaccines11081289_

Round 1

Reviewer 1 Report

This manuscript was described the relationship between COVID-19 vaccine and pregnant or lactating women. Generally, those women get nervous for medication or vaccines. For the widespread vaccination, it will be important to analyze the differences of their minds and policies between their backgrounds, such as races or countries. My questions and comments are following;

Major comments

1, Abstract

Authors described “ethnic minorities had a greater degree of vaccine acceptance”. It is very difficult to derive one conclusion about ethnic minorities, using only limited several publications, because there are many kinds of ethnic minorities with different characters in the world. Although it may be useful knowledge to show and discuss the results about ethnic minorities with their characters in the text, Abstract should be described by the universal results that were obtained by this study.

2, Figure 1

In Figure 1, 63 publications were selected finally. However, analyses in the text and Tables were performed by 21 publications. “included” in Figure 1 should be corrected from 63 to 21, because readers will be confused.

3, Results

In results, authors summarized selected publications in each topic. Majority of topics were summarized by text, without real data. Therefore, to understand the results correctly, we have to read both the text and tables, including supplementary tables. It will be helpful to understand, when real data in several publications are shown in one figure.

Minor comments

1, Table 2

Due to the wrong size, the right part of the table was cut off.

2, Table 3

The table was described three lines, which were black lines, gray lines and no lines. However, I could not understand the difference between three lines. Please make sure the table.

3, Numbering in Results

Numbering was wrong in Results. Results was started from “3.1 Literature search” and be back “3.1.1” after “3.4”. Then, it was numbering from “3.2” to “3.5” again after “3.1.5”. “3.2”, “3.3” and “3.4” were duplicated. Please make sure the numbering.

Author Response

thanks for your suggestions

Reviewer 2 Report

By addressing the following shortcomings and providing more detailed information, the manuscript can improve the overall comprehensiveness and impact of its analysis.

Abstract:
The abstract states that only observational, population-based studies were included. While this choice may have been deliberate, it limits the inclusion of other types of studies, such as qualitative research or randomized controlled trials, which could provide additional insights into vaccine hesitancy in pregnant and lactating women. Secondly, the abstract does not provide information on the sample sizes of the included studies. Without this information, it is challenging to assess the representativeness of the findings and the precision of the estimated vaccine hesitancy rates. Reporting the sample sizes would enhance the credibility of the analysis. Moreover, the abstract mentions that vaccine hesitancy rates ranged from 26% to 57% across different studies. This wide range suggests significant heterogeneity in the reported rates of vaccine hesitancy. It would be valuable to explore the reasons behind this variation and investigate potential factors contributing to the disparities observed across different populations. Lastly, the abstract briefly mentions that information gaps should be addressed to alleviate concerns related to adverse events. However, it does not provide any specific recommendations for healthcare providers, policymakers, or researchers. Including practical recommendations based on the findings would enhance the practical utility of the manuscript.

Introduction:

Introduction section needs to be rewritten and addressing the following shortcomings would improve the clarity, coherence, and depth of the introduction, providing a stronger foundation for the study and its objectives.

1)      The introduction lacks a clear and logical structure, making it difficult to follow the flow of ideas and arguments.

2)      The introduction does not adequately explain why vaccine acceptance and hesitancy among pregnant and breastfeeding women are particularly important or why they require special attention compared to other populations.

3)      While the introduction briefly mentions various factors influencing vaccine hesitancy, such as safety concerns and social media influence, it does not delve into a comprehensive discussion of these factors or provide a detailed understanding of their impact on pregnant and breastfeeding women.

4)      The research questions stated in the introduction are vague and do not clearly define the specific objectives of the study, making it challenging to understand the intended focus and outcomes. The research questions stated in the introduction are vague and do not clearly define the specific objectives of the study, making it challenging to understand the intended focus and outcomes.

5)      The introduction mentions that the current paper focuses on population-based studies, while hospital-based studies will be addressed separately. However, the rationale for this division and the potential impact on the overall findings are not discussed.

Methodology:

While the methodology mentions that the data extraction and risk of bias assessment were performed by two reviewers, it does not specify whether inter-rater reliability was assessed. The absence of inter-rater reliability assessment raises concerns about the consistency and reliability of the data extraction and risk of bias assessment, as discrepancies between reviewers could impact the accuracy and validity of the findings.

Results:

The exclusion criteria and reasons for excluding certain articles are not clearly explained. The manuscript mentions that articles were excluded based on duplicates, language, and title/abstract screening, but it does not provide specific details on how these decisions were made. This lack of transparency raises concerns about the potential for bias in the selection process. The sample sizes of the included studies vary widely, from 72 to 25,111 participants. This large variation in sample sizes can impact the generalizability and reliability of the findings. It would have been helpful to provide more information on the representativeness of the study populations and the statistical power of the included studies. Overall, these shortcomings in the literature search process, article selection, and reporting of sample characteristics limit the robustness and reliability of the findings presented in the manuscript.

Discussion:

The following discussion of the manuscript has several important shortcomings. Firstly, it fails to provide a clear and comprehensive analysis of the specific vaccines that have been studied in pregnant women. The discussion mentions "most available vaccines" without specifying which ones have been shown to be safe and effective during pregnancy. This lack of specificity limits the usefulness of the information provided. Secondly, the discussion focuses primarily on vaccine hesitancy without delving into the underlying reasons for this hesitancy or proposing concrete solutions. It mentions fear and lack of information as drivers of vaccine hesitancy but does not provide a thorough exploration of these factors or offer practical strategies for addressing them. Lastly, the discussion mentions the importance of proper communication between healthcare professionals and patients but does not elaborate on what effective communication entails or provide specific recommendations for improving communication skills. This omission hinders the practical applicability of the suggestions made.

There are a few instances of grammatical and language errors in the manuscript. Here are a few examples from discussion section:
"a part those" should be "apart from those" (line 347).
"it is enforced by healthcare providers’ uncertain regarding" should be "it is influenced by healthcare providers' uncertainty regarding" (line 352).
"considering that they have an increased risk of severe outcomes for several vaccine-preventable diseases" should be "considering that they are at an increased risk of severe outcomes for several vaccine-preventable diseases" (line 355).
"A higher level of education was correlated with greater vaccination acceptance" should be "A higher level of education correlated with greater vaccination acceptance" (line 392).
"communication techniques to medical and nursing staff should represent a top priority" should be "communication techniques to medical and nursing staff should be a top priority" (line 409).

These are just a few examples, and there are other grammatical and language issues throughout the manuscript. It would be advisable to thoroughly proofread and edit the entire manuscript to ensure clarity, coherence, and grammatical correctness.

Author Response

Thanks for your suggestions

Reviewer 3 Report

The authors have analysed the existing literature about anti-SARS-CoV-2 vaccine hesitancy in pregnant and lactating women. The study was conducted on PubMed/MEDLINE, ExcerptaMedica Database (EMBASE), and Scopus. The authors have included articles regarding the COVID-19 vaccine’s acceptance and/or refusal by pregnant and lactating women. The authors 21 papers were included in this study. All the included studies were cross-sectional and from Europe and North America. The authors have concluded that there is significant anti-SARS-CoV-2 vaccine hesitancy among pregnant women. The paper is generally well written and structured, but in my opinion, it has some shortcomings. My suggestions are as follows:

1.      Authors are advised to modify the title; the current title is so common. The present article is also a slightly modified version of a previously published article.

 Gianfredi, V.; Berti, A.; D’Amico, M.; De Lorenzo, V.; Castaldi, S. Knowledge, Attitudes, Behavior, Acceptance, and Hesitancy in Relation to the COVID-19 Vaccine among Pregnant and Breastfeeding Women: A Systematic Review Protocol Women 2023, 3, 73–81 https://doi.org/10.3390/women3010006.

2.      The willingness or unwillingness of pregnant women to receive the COVID-19 vaccination by site and the reasons for not wanting to be vaccinated should be included in the manuscript, along with the relative risk and 95% confidence intervals for being unwilling to be vaccinated by site and maternal characteristics.

3      3. Authors should ensure globally published literature is included to impact the scientific community in the current scenario.

  Authors should ensure that their article has been carefully checked for spelling, language, grammar, and style (where appropriate) throughout the manuscript.

I tried to correct the abstract.

The anti-SARS-CoV-2 vaccination is recommended for pregnant women due to the higher risk of complications. However, pregnancy has been associated with vaccine hesitancy. Our review aims at summarising the existing literature about anti-SARS-CoV-2 vaccine hesitancy in pregnant and lactating women. The research was conducted on PubMed/MEDLINE, ExcerptaMedica Database (EMBASE), and Scopus. Articles regarding the COVID-19 vaccine’s acceptance and/or refusal by pregnant and lactating women were selected. Only observational, population-based studies were included. The Joanna Briggs Institute's quality assessment tools were employed. A total of 496 articles were retrieved, and after the selection process, 21 papers were included in the current analysis. All the included studies were cross-sectional, mostly from Europe and North America. All of them included pregnant subjects, except one that focused on breastfeeding women only. Vaccine hesitancy rates ranged from 26% to 57% among different studies. Fear of adverse events and a lack of knowledge were shown to be the main drivers of hesitancy. Surprisingly, ethnic minorities had a greater degree of vaccine acceptance. Primigravidae were also shown to be more likely to accept the anti-SARS-CoV-2 vaccination. Our findings confirm significant anti-SARS-CoV-2 vaccine hesitancy among pregnant women. Information gaps should be addressed to contain concerns related to adverse events.

Author Response

Thanks for your suggestions

Reviewer 4 Report

The topic is of general interest, although there have been prior publications in this regard. As presented, there is a lot of repetition in terms of text and tables that need to addressed. The authors need to focus on the actual results.

Some specific recommendations;

Either delete Table 1 or the prior description in section 2.2

Delete section 3.1 as Figure 1 better illustrates the selection process

No need for supplemental Table 2

Delete Tables 2 and 3

Delete the text in section 3.4

Include supplemental Table 5 in the main presentation.

Some adjectives are not necessary; eg Surprisingly...

Keep the descriptions factual and without commentary adjectives

Author Response

Thanks for your suggestions

Round 2

Reviewer 1 Report

Table 3 was not corrected. Please ask the editorial office, if authors cannot correct. I do not have additional comments.

Author Response

Table 3 was not corrected. Please ask the editorial office, if authors cannot correct. I do not have additional comments.

We thank the Reviewer for having raised this point. We cannot understand what is the problem since we correct visualize Table 3. Moreover, this point was not raised by the other Reviewers. We hope this aspect can be solved during the publication phase.

Reviewer 2 Report

Abstract: The abstract provides a concise overview of the study. However, it lacks specific details about the methodology, results, and conclusions, making it less informative for readers who want a quick understanding of the paper's content.

Introduction: The introduction is quite lengthy and contains unnecessary repetitions and excessive background information. It should be more focused on introducing the concept of vaccine hesitancy among pregnant and lactating women and how it pertains to the current study.

Materials and Methods: This section is comprehensive, but it can be more concise. It should provide a clear outline of the study design, data sources, inclusion/exclusion criteria, and quality assessment process without excessive details.

Results: The results section is overly detailed and includes unnecessary information. It should focus on presenting key findings, such as vaccine hesitancy rates, main drivers, and demographic characteristics of the studied population.

Discussion: The discussion section is also lengthy and repetitive. It should be restructured to focus on the implications of the findings, potential limitations, and future research directions.

Conclusion: The conclusion should be a succinct summary of the main findings and their significance.

Overall, the manuscript contains valuable information, but it needs to be significantly shortened and reorganized to improve its clarity and readability. Reducing unnecessary details and repetitions will make it more suitable for publication. Additionally, adding specific information to the abstract and conclusion will enhance the paper's overall impact.

Author Response

1. Abstract: The abstract provides a concise overview of the study. However, it lacks specific details about the methodology, results, and conclusions, making it less informative for readers who want a quick understanding of the paper's content.

a) Dear Reviewer, thank you for this comment. Regarding the methodology details, the abstract contains information about databases searched, inclusion criteria, tool used for the quality evaluation. The following sentences are part of the abstract: “The research was conducted on PubMed/MEDLINE,ExcerptaMedica Database (EMBASE) and Scopus. Articles regarding the COVID-19 vaccine’s acceptance and/or refusal by pregnant and lactating women were selected. Only observational, population-based studies were included. The Joanna Briggs Institute quality assessment tools were employed.” Moreover, in order to
meet reviewer’s request, we added the information about PRISMA guidelines.

b) Regarding results, the abstract contains information on total amount of articles retrieved, number of included studies, main characteristics of included studies, sample size, vaccine hesitancy rate and main driver for vaccine hesitancy. Lastly, abstract also contains information regarding quality evaluation results. The following sentences are part of the results section of the abstract: “A total of 496 articles were retrieved, and after selection process 21 papers were included in the current analysis. All the included studies were crosssectional, mostly from Europe and North-America. The sample sizes ranged between 72 to 25111 subjects. All of them included pregnant subjects, except one that focused on breastfeeding women only. Vaccine hesitancy rates ranged from 26% to 57% among different studies. Fear of adverse events and lack of knowledge were shown to be the main drivers of hesitancy. Approximately half of the studies (11/21) were classified as low quality, the remaining (9/21) were classified as moderate, and only one study was classified as high quality Primigravidae were also shown to be more likely to accept an-ti-SARS-CoV-2 vaccination.”

c) Regarding conclusions, we stated the following: “Our findings confirm significant anti-SARS-CoV-2 vaccine hesitancy among pregnant women. Information gaps should be addressed to contain concerns related to adverse events.”

d) We hope this can meet Reviewer request. Otherwise, we are glade to modify our abstract according to specific Reviewer’s requests.

2. Introduction: The introduction is quite lengthy and contains unnecessary repetitions and excessive background information. It should be more focused on introducing the concept ofvaccine hesitancy among pregnant and lactating women and how it pertains to the current study.

a) We thank the Reviewer for this suggestion. The following sentences have been removed: “It should be considered that vaccine hesitancy is often a vaccine-specific phenomenon and depends on the socio-cultural background of the hesitant subject. More-over, COVID-19 vaccines are far more recent than any other immunization product, making it difficult to foresee the evolution of hesitation towards them. We therefore designed a systematic review and meta-analysis protocol with the purpose of better understanding the underlying
mechanisms for COVID-19 vaccine hesitancy among pregnant/breastfeeding women.”.

b) Regarding the focus on vaccine hesitancy, we reported the following: “Vaccine acceptance is influenced by multiple determinants dynamically interacting with each other, including the subject’s knowledge and attitudes regarding vaccination, societal norms, perceived vaccination-related benefits and risks [1]. In case of insufficient drive towards vaccination, the subject might hesitate or even refuse vaccination. This wide spectrum of behaviors leading to delays in vaccination schedules’ completion is currently defined as vaccine hesitancy [2]. Hesitancy may arise from multiple factors, including concerns about the
product’s safety and effectiveness, mis-trust in either vaccine development or regulatory processes. These attitudes and beliefs may stem from various causes, among which misinformation is one of the main determinants [3, 4]. Vaccine hesitancy has been listed as one of the top ten threats to global health by the World Health Organization (WHO) due to its impact on vaccina-tion coverage, leading to a decrease in immunization rates [5].”

c) Regarding vaccine hesitancy among pregnant and lactating women, we reported the following: “The complexity of these phenomena increases when dealing with pregnant/breastfeeding women [6]. In fact, during pregnancy and breastfeeding, women often seek information regarding their child’s health as well as their own, which is expected to influence significantly the subject’s behavior towards medical matters [7]. At the same time, these women are highly recommended to promptly undergo vaccination, as various vaccine preventable diseases have a significantly higher chance of causing severe outcomes during pregnancy or might be spread to the child either before or after birth.”

d) In the last section of the introduction, we related vaccine hesitancy among pregnant/lactating women towards the COVID-19 vaccine and concluded our introduction detailing research questions and aims of the current study.

e) A few changes have been made to the paragraphs’ structure, in order to help the reader focus their attention on the paper’s main topics.

f) We hope this can be considerate appropriate.

3. Materials and Methods: This section is comprehensive, but it can be more concise. It should provide a clear outline of the study design, data sources, inclusion/exclusion criteria, and quality assessment process without excessive details.

a) We thank the Reviewer for having highlighted this point. We shortened the methods section. We hope this can be considered adequate by the Reviewer.

4. Results: The results section is overly detailed and includes unnecessary information. It should focus on presenting key findings, such as vaccine hesitancy rates, main drivers, and demographic characteristics of the studied population.

a) We thank the Reviewer for this consideration. However, the aim of the review was to synthesize the existing literature on knowledge, beliefs, attitudes, barriers and facilitators relating to acceptance/refusal of COVID-19 vaccination among pregnant and breastfeeding women. Therefore, we cannot only focus on vaccine hesitancy. This will result in a reporting selection, which will represent a deviation from the review’s protocol. We hope this decision could be appreciated by the Reviewer.

5. Discussion: The discussion section is also lengthy and repetitive. It should be restructured to focus on the implications of the findings, potential limitations, and future research directions.

a) We thank the Reviewer for this comment. We largely modified the discussion section. Moreover, a paragraph 4.1 Implication for policies and practice has been added to the manuscript.

6. Conclusion: The conclusion should be a succinct summary of the main findings and their significance.

a) We thank the Reviewer for having highlighted this aspect. The conclusions have been modified as follow: “To conclude, this review summarized data from 21 cross-sectional studies focusing on pregnant and breastfeeding women aged 15-55 years, with a good geographical representativeness. The positive attitude toward COVID-19 vaccine ranged between 29.7% and 80% and several factors were considered as predictors. Among them, higher educational level, being primigravida, having received previous vaccinations or vaccines recommendations from health care professionals were associated with a higher
attitude, while results on socioeconomic status are discordant. COVID-19 vaccine
acceptance ranged between 20% and 68%. In this case, older age, ethnic minorities, higher income and education, having received previous vaccines or vaccines’ recommendations were all statistically significant associated with higher acceptance. Despite available evidence of safety and effectiveness of the anti-SARS-CoV-2 vaccination in pregnant women, hesitancy still remains, thus threatening vaccination campaigns’ efficacy all over the world. According to our review, COVID-19 vaccine hesitancy ranged between 26% to 57%. Vaccine hesitancy’s main drivers appear to be fear and lack of information, proving
that proper communication between healthcare professionals and their patients is the best way to counter uncertainties and doubts related to vaccination. Improving knowledge and awareness is the first point to increase vaccine confidence, and the failure to adequately inform pregnant women exasperates the skepticism about vaccination. An effort should be made to invest in communication-related soft skills of healthcare professionals, while also
reinforcing mass communication via both new and traditional media.”

7. Overall, the manuscript contains valuable information, but it needs to be significantly shortened and reorganized to improve its clarity and readability. Reducing unnecessary details and repetitions will make it more suitable for publication. Additionally, adding specific information to the abstract and conclusion will enhance the paper's overall impact.

a. We thank the Reviewer for the appreciation and for valuable suggestions. We hope the revised version of the manuscript is now suitable for publication.

Reviewer 4 Report

The Authors have responded to the comments.

Author Response

We thank the Reviewer for having provided valuable comments, helping us to improve the overall quality of the manuscript.

Round 3

Reviewer 2 Report

The manuscript is now in better form for publication